



# Evaluation of vertically resolved longwave radiation in SPARTACUS-Surface 0.7.3 and the sensitivity to urban surface temperatures

Megan A. Stretton[1], William Morrison[1,2], Robin J. Hogan[1,3], Sue Grimmond[1],

[1]Department of Meteorology, University of Reading, Reading, UK
[2]Chair of Environmental Meteorology, Faculty of Environment and Natural Resources, University of Freiburg, Freiburg, Germany
[3]European Centre for Medium-Range Weather Forecasts, Reading, UK

*Correspondence to*: Megan A. Stretton (m.stretton@pgr.reading.ac.uk)

**Abstract.** Cities materials and urban form impact radiative exchanges, and hence both surface and air temperatures. Here, the 'SPARTACUS' multi-layer approach to modelling longwave radiation in urban areas (SPARTACUS-Urban) is evaluated using the explicit DART (Discrete Anisotropic Radiative Transfer) model. SPARTACUS-Urban describes realistic 3D urban geometry statistically, rather than assuming an infinite street canyon. Longwave flux profiles are compared across an August day for a 2 km x 2 km domain in central London. Simulations are conducted with multiple temperature

configurations, including realistic temperature profiles derived from thermal camera observations. The SPARTACUS-Urban model performs well (cf. DART) when all facets are prescribed a single temperature, with normalised bias errors (nBE) < 2.5% for longwave downwelling at the surface, and < 0.5% for the upwelling longwave at the top of the canopy. Errors are larger (nBE < 8%) for the net longwave fluxes from walls and roofs. Using more realistic surface temperatures, which vary depending on whether a surface is sunlit, the nBE in upwelling longwave increases to ~2%. Errors in roof and wall net

longwave fluxes increase through the day, but still nBE are 8–11%. This increase in nBE occurs because SPARTACUS-Urban represents vertical variation of surface temperature but not horizontal variations within a domain. We conclude that SPARTACUS-Urban accurately predicts longwave fluxes, requiring less computational time cf. DART, but with larger errors when surface temperatures vary because of being sunlit and/or shaded. SPARTACUS-Urban could enhance multi-layer urban energy balance schemes prediction of within-canopy temperatures and fluxes.

## 1    Introduction

The differences in energy exchanges between urban and rural areas leads to canopy layer air temperature differences of 3-10°C (Oke 1987). This phenomenon, known as the canopy layer urban heat island effect (CL-UHI), has been studied and observed worldwide (Oke 1982; Zhang et al. 2012; Wu et al. 2014; Guo et al. 2016; Dou and Miao 2017; Gaitani et al. 2017). The CL-UHI is driven by contrasting radiative exchanges between urban and rural environments, resulting from the

heterogeneous nature of cities (Aida and Gotoh 1982; Oke 1982; Kondo et al. 2001; Harman and Belcher 2006; Ao et al. 2016). With increasing urbanization, and more people residing in cities than rural areas since 2007 (Heaviside et al. 2017),



there is greater exposure of vulnerable people to extreme weather, such as heatwaves, with the severity of such events potentially exacerbated by the CL-UHI.

The heterogenous 3D structures of urban areas lead to changes in the surface energy balance, and diurnal temperatures (Souch and Grimmond 2006; Masson et al. 2008), due to the resultant differential shortwave (SW) input and radiative cooling across a city. The crenulated urban morphology and resultant deep canyons cause an uneven exposure to the sky and an increased the surface area available for exchange (cf. rural areas), which increases the SW absorption throughout the day. This differential solar irradiance drives temperature variations between facets, including vertical gradients (Oke 1981;
Blankenstein and Kuttler 2004; Harman and Belcher 2006; Hénon et al. 2012; Hu and Wendel 2019).

The spatial variation of facet temperatures is highest during the daytime, due to variations in the absorption and reflection of the dynamic solar radiation (Myint et al. 2013; Crum and Jenerette 2017; Antoniou et al. 2019). However, temperatures remain high overnight from the morphology reducing exposure to the sky therefore increasing radiative trapping and slowing
cooling rates, and lower effective albedo. Facet materials (e.g., concrete, tarmac) can have low albedo, high heat capacities and high thermal inertia (Bohnenstengel et al. 2011). This results in large daytime heat storage in to urban volume, which is released slowly at night (Meyn and Oke 2009; Kershaw and Millward 2012).

These impacts on the radiative and other energy exchanges needs to be parameterised within by the numerical weather
prediction (NWP) land surface schemes (Masson 2006). A common approach to simplifying the 3D structure of cities is to treat the urban form as a canyon between buildings of equal height (Nunez and Oke 1977). Initially, in some standalone models, some complexity was considered (e.g., allowing intersections) (e.g., Aida 1982; Arnfield 1982a, 1988), when modelling urban radiative exchanges. But for NWP, with computer resource limitations, this was simplified to assuming an infinite canyon, as it simplifies the view factor geometry and computations (e.g., Masson 2000; Harman et al. 2004), which
is also used for the other energy balance fluxes (e.g., Masson 2000; Kusaka et al. 2001a; Lee and Park 2008). Many of these models calculate the fluxes for individual facets (wall, roof, and ground) (Masson 2006). However, assuming the constant building height and lack of intersections neglects the variability of urban geometry (e.g., clusters of tall buildings, courtyards) that influence shadowing and trapping of radiation, and wind fields (e.g., Hertwig et al. 2019, 2021).

Sub-facet differences (e.g., roof orientation, and slopes, high/low parts of walls, wall orientation, sunlit/shaded pavement) can create surface temperature variability, which is not captured if represented by a single mean surface temperature in an urban energy balance scheme (Hilland and Voogt 2020). For example, diurnal variations of wall temperature are linked to their orientation relative to the sun, and additionally to inter-building interactions (e.g. shadows) (Nazarian and Kleissl 2015; Antoniou et al. 2019). This is important as 12-50% of the urban surface is comprised of walls (Voogt et al. 1997; Grimmond
and Oke 1999; Hénon et al. 2012). Similarly, roofs differ from walls, with high incident SW radiation (Harman and Belcher



2006; Morrison et al. 2018), and ground surfaces in deep urban canyons may have dampened diurnal temperature variability (Hu and Wendel 2019). Inclusion of the vertical variability of the urban form may allow such features to be captured by models, unlike within the infinite homogenous canyon approach.

Some of these features can be addressed by multi-layer radiative transfer models (e.g., Seoul National University Canopy Model (Ryu and Baik 2012; Ryu et al. 2013), and SPARTACUS-Urban (Hogan 2019)), which allow for variations in roof and wall heights. This leads to more nuanced radiative trapping and more realistic vertical temperature distributions. The SPARTACUS-Urban model uses vertical profiles of urban geometry to simulate radiation between buildings to account for atmospheric absorption, emission, and scattering, whilst being fast enough to be used for NWP. Wall and roof areas are
derived from building footprint data but can be simplified to two parameters. The shortwave (SW) capabilities, evaluated using an explicit radiative transfer model, are in good agreement for realistic urban domains (Stretton et al. 2022).

In this study, the longwave (LW) capabilities of SPARTACUS-Urban are evaluated for the first time, using both DART (Discrete Anisotropic Radiative Transfer (Gastellu-Etchegorry et al. 2015) and the Harman et al. (2004) models (Sect. 2).
The former is an explicit scheme. The latter takes a common approach used in operational weather and climate models. We examine SPARTACUS-Urban's prediction of LW fluxes for a domain in central London, with varying complexities of facet temperatures derived from thermal camera observations (Morrison et al. 2020, 2021) to undertake the evaluation (Sect. 3). A comparison of SPARTACUS-Urban and DART is made (Sect. 4) and with the Harman et al. (2004) street canyon radiation (Sect. 5). The results of the evaluation are presented (Sect. 6).

## 2   Radiative transfer models

### 2.1  SPARTACUS-Urban

The SPARTACUS approach, developed to model radiative exchange within cloud fields (Hogan et al. 2016), has been applied to both vegetated (Hogan et al. 2018) and built areas (Hogan 2019b). Obstacles are assumed to be randomly distributed within the horizontal plane to radiation, allowing simulation of a mean radiation field with height. Although
SPARTACUS-Surface open-source software (Hogan 2021) includes both SPARTACUS-Urban and SPARTACUS-Vegetation, given our buildings focus (i.e., excluding urban vegetation), we refer to this as "SPARTACUS-Urban". We previously used DART to evaluate SW part of SPARTACUS-Urban for multiple urban geometry configurations (Stretton et al. 2022).

A discrete-ordinate method is used to solve coupled ordinary-differential equations for $2N$ radiation streams ($N$ streams per hemisphere, here $N = 8$). Radiative fluxes are calculated per height interval, $z$, for layers split into clear-air and building 'regions' in the horizontal plane. The incoming and outgoing fluxes (W m$^{-2}$), and absorption (W m$^{-3}$) profiles are calculated



for three facets (wall, roof, and ground). SPARTACUS-Urban requires profiles information for each scene (i.e., geometry, emissivity (ε), and surface temperature $T$) provided as plan area fraction ($\lambda_p$), building edge length ($L$), with height, $z$. These, and other, morphology parameters can be derived from building footprint data (Martilli 2009; Kent et al. 2019; Stretton et al. 2022). SPARTACUS-Urban allows vertical variation of facet temperatures to be prescribed with one facet $T$ per height level. Although we assume a vacuum, SPARTACUS-Urban can account for atmospheric absorption. For this paper, we assume a wavelength of 10 μm (where atmospheric absorption is weak), so the emission rate in SPARTACUS-Urban (and DART) make use of the Plank function at 10 μm, with a top-of-canopy downwelling longwave spectral flux at that wavelength (LW↓).

## 2.2 DART

The DART (Discrete Anisotropic Radiative Transfer) model (Gastellu-Etchegorry et al. 2015) simulates the exchanges of radiation in heterogenous scenes. Scenes are imported as 3D models which can contain vegetation and buildings (simulated as turbid media and planar triangle facets, respectively), with varying topography and a within-canopy atmosphere. These scene elements interact with radiation iteratively within a 3D array of voxels. Scene elements within a given voxel can interact with each other, and per-voxel fluxes are stored after each iteration. To model the LW exchanges in DART, both a 3D building model and a full 3D field of surface temperatures (prescribed to the triangles) are needed. The latter can be determined by solar irradiance. DART's planar triangles can only have one $T$, therefore if a single triangle covers the full vertical extent of a building facet, it will only have one temperature to describe the vertical variation.

DART has been evaluated using observations for vegetation (Sobrino et al. 2011) and relative to other models (Widlowski et al. 2015). It has been applied in urban areas (e.g., Landier et al. 2018; Chrysoulakis et al. 2018; Morrison et al. 2020) and used for evaluations of simpler radiative transfer models (e.g., SPARTACUS-Urban, Stretton et al. (2022)).

## 2.3 Single-layer street canyon

The Harman et al. (2004) approach has a system of linear equations that compute the exact radiative transfer with vertically constant wall temperatures. Hogan (2019b) compared SPARTACUS-Urban to Harman et al.'s (2004) LW radiation after modifying Harman's horizontal geometry to have an exponential distribution, as assumed by SPARTACUS, but with all buildings having equal height. Agreement was found between the two models for the net outward LW flux from the ground and walls when greater than 4 streams were used by SPARTACUS Here, we use the Harman implementation in the SPARTACUS-Surface software package (implemented as in Sect. 4.2 of Hogan (2019)).

Harman assumes two parallel buildings of infinite length with constant height, $H$, separated by a constant street width, $W$. For this comparison, the total area of the ground, walls, and roofs are equal to the real-world domain (Sect. 3.1) with $H$ set



equal to the mean building height, $\overline{H}$, and the building fraction equal to the surface ($\lambda_p(z = 0)$) value. The *H/W* is calculated

using (Hertwig et al. (2020) their Eq. 3):

$$\frac{H}{W} = \frac{\pi}{2} \frac{\lambda_f}{\left(1 - \lambda_p\right)} \tag{1}$$

where the frontal area index, $\lambda_f$ is calculated using $\lambda_w = \lambda_f \pi$, where $\lambda_w$ is the total normalised wall area, calculated from the

vertical profile of normalised building edge length, *L*:

$$\lambda_W = \sum_i^n L_i \Delta z_i \,. \tag{2}$$

This implementation of Harman requires a single temperature (i.e., not a profile) for each of the three urban facets.

## 3   Methodology

### 3.1   Model Domain

Our evaluation focuses on a 2 km x 2 km area of central London (Figure 1a). This domain's buildings have varying

horizontal size and height, with some high-rise buildings used for residential and commercial purposes. The digital surface

model (DSM) and digital elevation model (DEM) used are derived from "Virtual London" building footprint dataset (Evans

et al. 2006). For heights, the 25[th] percentile of the DEM and 75[th] percentile of the DSM are used giving individual buildings

a single height, flat roofs, and flat, vertical walls.

For SPARTACUS-Urban, the required vertical profiles of $\lambda_p$ and *L* are derived from a 1 m x 1 m building footprint raster, as

rasterization removes internal walls between buildings. For DART, the 3D building roof and ground level geometry are

determined from the DSM and DEM. The Stretton et al. (2022) 3D building model is improved slightly (e.g. shift in vertical

plane, removal of some internal walls). The voxel resolution used in DART is 1 m vertically and 5.206 m horizontally.

SPARTACUS-Urban has the same vertical resolution as DART.



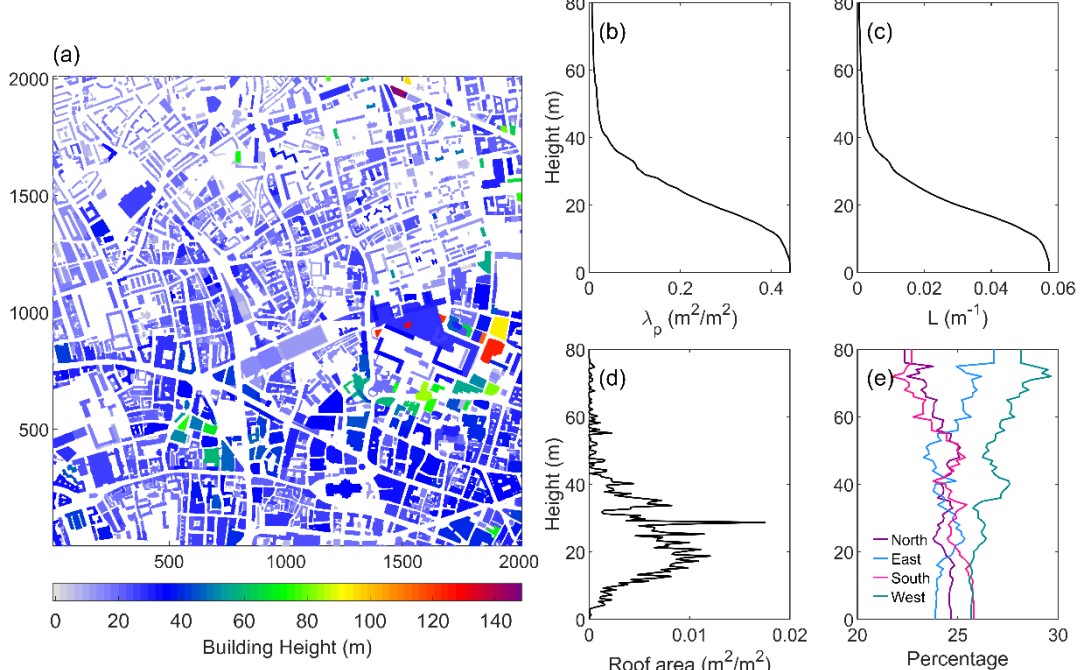

**Figure 1: Low level of detail central London domain (i.e., flat roofs): (a) building heights, (b) building plan area fraction ($\lambda_p$) with height, (c) normalised building edge length (L) with height (Eq. 2), (d) roof area with height, (e) wall orientation distributions calculated from surface-classified DART emission output.**

### 3.2 Observations used for radiative transfer inputs

In the model domain, three observation sites are present (Table 1). We focus on a day (27[th] August 2017) with detailed surface temperature observations and almost clear skies (< 45-min cloud mid-afternoon) (Morrison et al. 2020).

Given computational constraints, DART is run for a single wavelength (10 μm). Hence, we are unable to use broadband longwave flux measurements. Instead, we have rerun the ECMWF atmospheric radiation scheme using pressure, temperature and humidity profiles from ERA5 (Hersbach et al. 2020) for that day (Figure 2), and extracted bottom-of-atmosphere (BOA) clear-sky downwelling spectral flux at 10 μm. For the SPARTACUS-Urban and Harman et al. (2004) simulations, changes were made to SPARTACUS-Surface so that emission is calculated for a single spectral wavelength. SPARTACUS-Surface additionally requires $T_{Air}$, but as we simulate radiative fluxes in a vacuum, this is set to 0 K. Each model requires an emissivity, $\varepsilon$, for each surface. We use a homogenous $\varepsilon = 0.93$, based on the mean urban value determined in the Kotthaus et al. (2014b) spectral library.



Facet surface temperatures are prescribed using thermal camera imagery (Optris PI-160 LW infrared cameras) observed for a 420 m x 420 m area within this domain (Morrison et al. 2020, 2021) (Figure 3). Detailed modelling has categorised these observations by facet type, sunlit/shaded, and orientation (Morrison et al. 2020, 2021). Surface temperatures are split into roof, ground, and cardinal wall orientation (*etc.*) types. Although we evaluate SPARTACUS-Urban across the whole day, to

demonstrate the performance for multiple surface temperature configurations, we select times with distinct temperature profiles (e.g., just after sunrise, with no facet temperature range) and summarise the general model performance. As surface temperature processing constraints (Morrison et al. 2020) gives observations from 5:45 UTC (sunrise: 5:04 UTC), the models are runs for every hour from then to end of the day. The mid-afternoon cloud period is discarded, as no sunlit/shaded temperature range is observed (Figure 3).


**Table 1: Sensors used from within domain (Figure 1a). Meteorological time series, and further details of observations within this domain can be found in Morrison et al. (2021)**

| Site | Full name | Latitude °N | Longitude °W | Instruments |
|---|---|---|---|---|
| BCT | Barbican Cromwell Tower | 51.5206 | 0.09230 | Davis weather station |
| IMU | Islington Michael Cliffe House Upper | 51.526 | 0.1061 | Davis weather station<br>Kipp and Zonen CNR1 radiometer<br>Optris Pi160 infrared thermal camera |
| WCT | Wycliffe Court Tower | 51.5267 | 0.1036 | Optris Pi160 infrared thermal camera |



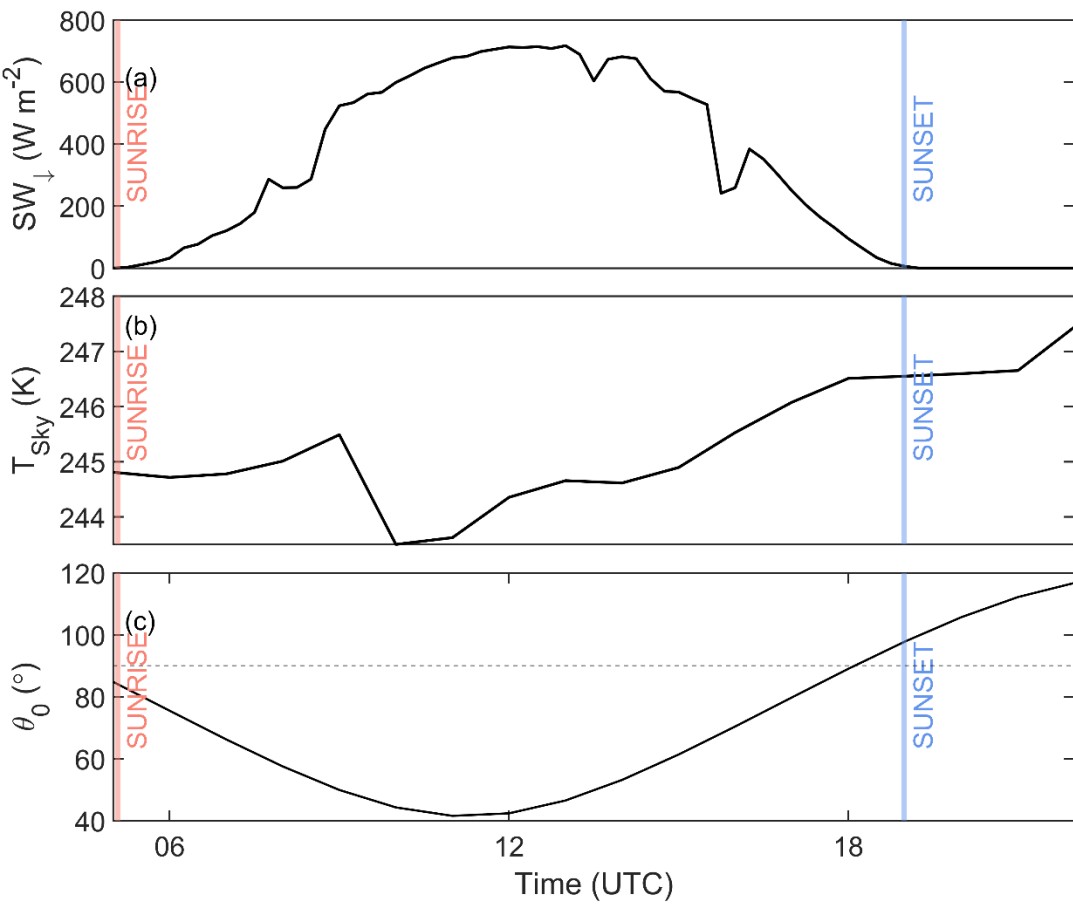

**Figure 2: Diurnal timeseries of (a) downwelling shortwave (SW↓) observations from a Kipp and Zonen CNR1 radiometer located at IMU, (b) Clear-sky 10-mm brightness temperatures calculated from ERA5, and (c) solar zenith angle ($\theta_0$). Additional meteorological observations for the day of interest are shown in Morrison et al. (2021).**



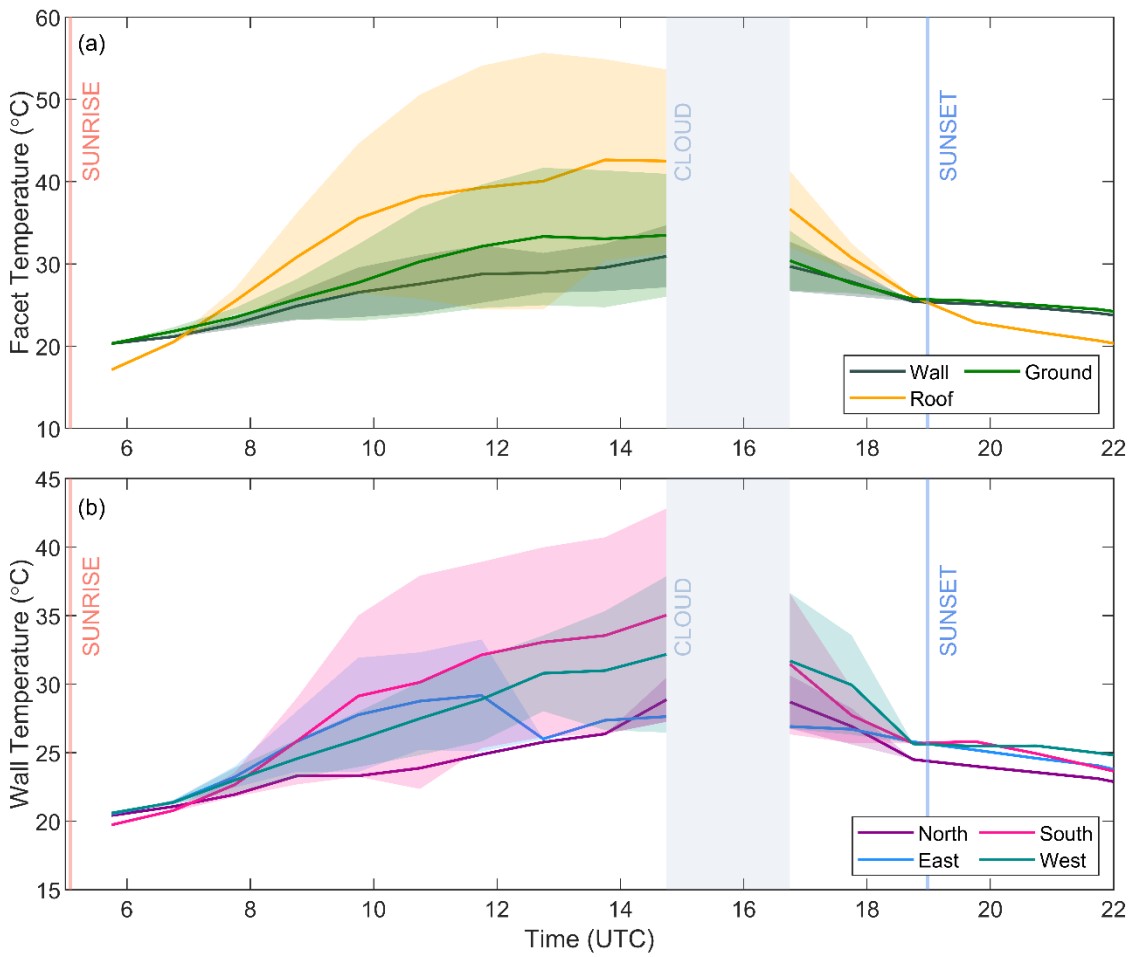

**Figure 3: Mean (line) and range (shading; mean [sunlit to shaded]) temperature observations (Morrison et al. 2021) for each (a) facet type (walls - all weighted equally) and (b) wall azimuthal orientation.**

### 3.3 Model surface temperature (*T*) prescription

The three radiative transfer models (Sect. 2) require different *T* inputs. To assess the sources of error between SPARTACUS-Urban and DART (i.e., radiation calculation or surface temperature values), two complexities of model runs are undertaken. First, simulations assume an isothermal temperature within each surface type, with DART surfaces prescribed the single mean *T* from the camera observations (Figure 3a, line). To match this, SPARTACUS-Urban roofs and ground are prescribed the mean DART input temperature. For SPARTACUS-Urban $T_{Wall}$, each wall orientation is weighted equally (Figure SM 1), following the SPARTACUS-Urban assumption that walls equally face in all directions, such that:

$$T_{Wall} = (T_{Wall\,N} + T_{Wall\,E} + T_{Wall\,S} + T_{Wall\,W})/4 \qquad (3)$$

where $T_{,Wall\,N\text{-}E\text{-}S\text{-}W}$ are one of the four cardinal directions. For Harman, the same temperatures as SPARTACUS-Urban are





used.

Non-isothermal surface temperatures varying by sunlit – shaded status allow for horizontal and vertically differences by facet type. These can be represented in multi-layer energy exchange schemes. DART scenes can have a temperature mean and range (Fig. 3) prescribed across a surface type (e.g., roof, west facing wall, east facing wall). This allows for the impacts of variation in SW flux to be simulated. A DART SW simulation is used to determine whether a triangle is sunlit or in shade,

and therefore which temperature to utilize.

To enable SPARTACUS-Urban to capture the horizontal surface temperature variations, SW SPARTACUS-Urban simulations are conducted to estimate the fraction of sunlit walls or roofs at each height interval ($F_{Sun,Wall,i}$, $F_{Sun,Roof,i}$, $F_{Sun,Ground}$), using the solar zenith angle ($\theta_0$), with shaded fractions determined by the difference e.g., $F_{Sh,Wall,i} = 1 -$

$F_{Sun,Wall,i}$. The sunlit and shaded roof temperatures ($T_{Sun,Roof}$, $T_{Sh,Roof}$) are simply weighted at each height by the sunlit and shaded fractions to obtain $T_{Roof,i}$. The ground temperatures ($T_{Ground,sun}$, $T_{Ground,sh}$) are treated in the same way, but at just $z=0$.

As the four wall orientations have different temperatures depending on their shadow history (Morrison et al. 2021), for SPARTACUS-Urban we weight them to obtain one average sunlit and shaded wall temperature ($T_{Wall,sun}$, $T_{Wall,sh}$). Given the

SPARTACUS-Urban assumption that walls face equally in all directions, we weight the sunlit and shaded temperatures (as Eq. 3), but use the solar azimuth angle, $\Omega$, to determine the 'dominant' sunlit wall orientation. The dominant sunlit facing surface (e.g., south) temperature (in this example, $T_{Sun,South}$) is double weighted in Eq. 3 (i.e., replacing $T_{Sun,North}$) assuming the wall 180° (i.e., north facing surfaces in example) are shaded. The opposite is done for the $T_{,Sh,Wall}$, obtaining (for this example):

$$T_{Sun,Wall} = 0.0 \cdot T_{Sun,Wall\ N} + 0.25 \cdot T_{Sun,Wall\ E} + 0.25 \cdot T_{Sun,Wall\ W} + 0.5 \cdot T_{Sun,Wall\ S},\qquad(4)$$

$$T_{Sh,Wall} = 0.5 \cdot T_{Sh,Wall\ N} + 0.25 \cdot T_{Sh,Wall\ E} + 0.25 \cdot T_{Sh,Wall\ W} + 0.0 \cdot T_{Sh,Wall\ S}.\qquad(5)$$

The $T_{Wall,sh}$ and $T_{Wall,sun}$ are weighted using $F_{Sun,Wall,i}$ and $F_{Sh,Wall,i}$ to determine the $T_{Wall,i}$ for each height:

$$T_{Wall,i} = F_{Wall,sun,i} T_{Wall,sun} + F_{Wall,sh,i} T_{Wall,sh}.\qquad(6)$$

To visualise this at several times see Figure SM 1. Combining $F_{Sun,Wall,i}$ and $F_{Sun,Roof,i}$ gives a larger weight to warmer sunlit

surface temperatures in the simulations, better matching the emission from the DART model scenes.

For the Harman et al. (2004) simulations, area-weighted surface temperatures from the SPARTACUS-Urban profiles are used:

$$T_{Wall} = \sum_{i}^{n} T_{Wall,i} \left( \lambda_{Wall,i} / \lambda_{Wall} \right)\qquad(7)$$

where $\lambda_{Wall,i}$ is the exposed normalised wall area at each height, which is normalised by the total wall area, $\lambda_W$. Eq. 7 is also

applied to roofs. This ensures that warmer surfaces at the top of the canopy with small areas are not overweighted.



### 3.4 Evaluation Metrics

We evaluate SPARTACUS-Urban using DART by comparing the profiles of LW upwelling and downwelling clear-air spectral fluxes ($LW_↑$, $LW_↓$), and the intercepted, outgoing, and net ( = incoming – outgoing, relevant for facet temperature evolution) flux into walls, roofs, and ground (i.e., $LW_{In,Wall}$, $LW_{Out,Wall}$, $LW^*_{Wall}$). The LW clear-air fluxes have units of W m$^{-2}$ µm$^{-1}$ of the entire horizontal scene, while the fluxes from walls and roofs have units W m$^{-3}$ µm$^{-1}$, as we divide by the layer thickness (1 m) to obtain a resolution independent flux.

For the comparison between SPARTACUS-Urban and DART, we examine the downwelling longwave at the base of the canopy, and the upwelling longwave at the top of the canopy in DART ($H_{max}$) to obtain a normalised bias error. The $LW_↑$ flux profiles are evaluated using the normalized bias error at a specified height ($nBE$), expressed as a percentage of the DART flux:

$$nBE = \frac{LW_{SU} - LW_{DART}}{LW_{DART}} 100\%. \tag{8}$$

We compare the differences in the wall and roof fluxes between the two models by using a nBE in the total interception, emission, and net LW flux, calculated from 1 m to $H_{max}$.

## 4   Results

### 4.1 Prescribed surface temperatures

The $F_{Sun,Wall,i}$ and $F_{Sun,Roof,i}$ are calculated from SPARTACUS-Urban SW simulations for each time period (Figure 4). The sunlit fraction into the canopy increases as zenith angle, $\theta_0$, decreases until about 11:45 UTC (Figure 2). As more walls become illuminated within the canopy, there is an increase in $T_{Wall}$ (Figure 3, Figure 4). As $\theta_0$ increases again (Figure 2c), the within-canopy surfaces become more shaded than sunlit.

From combining the $F_{Sh}$ and $F_{Sun}$ profiles with the DART prescribed facet $T$ (Eq. 4 to 6) the $T_{Wall}$ and $T_{Roof}$ profiles are obtained (Figure 5). At 5:45 UTC all DART temperatures are the same, so all temperature configurations and SPARTACUS-Urban temperatures are equal. At 7:45 UTC, the first vertical variations in temperature occur with sunlit roof facets higher in the canopy causing warmer temperatures above. Both 11:45 UTC and 13:45 UTC share similar $T_{Wall}$ profiles, and do not have much influence from the warmer south facing walls despite their greater weighting. The most different $T_{Roof}$ profile, spanning the widest temperature range, occurs at 17:45 UTC.



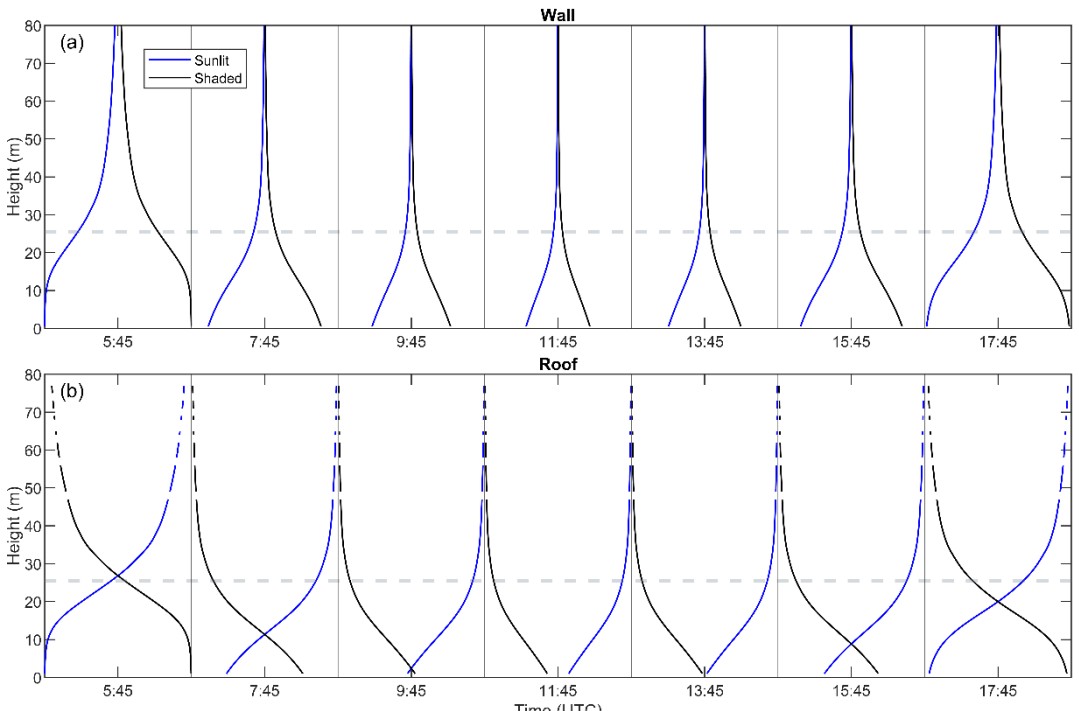

**Figure 4: Sunlit (blue) and shaded (black) fraction of (a) walls and (b) roofs during the study day from SPARTACUS-Urban shortwave simulations using solar zenith angles (Figure 2). Lines are shown as dashed when no roofs occurs at a height. Mean building height ($\bar{H}$ = 25.5 m, grey dashed).**

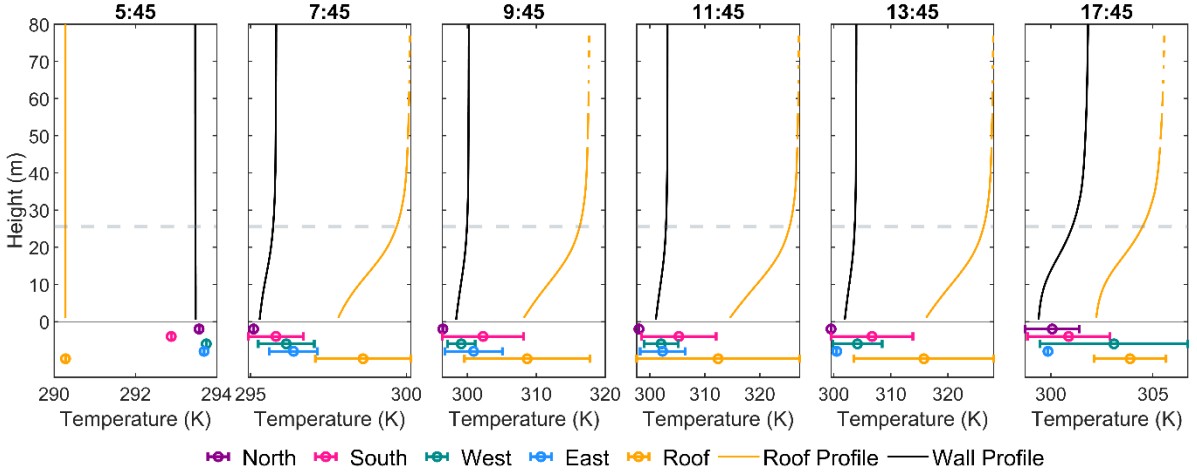

**Figure 5: Temperature profiles at six times (UTC) used in SPARTACUS-Urban simulations (averaging methods, Sect. 3.3) with temperatures prescribed to DART surface types given in the error bars below each set of temperature profiles, with the mean temperature denoted by open circles, and sunlit-shaded range given (Figure 3). Note x-axes differ between panels.**





### 4.2 Comparison of SPARTACUS-Urban and DART: One facet temperature ($T$)

First, when $T$ does not vary there is good agreement between SPARTACUS-Urban and DART. There is good agreement for LW$_\uparrow$ at the top of the canopy (nBE < 0.5% across the whole day, Table 2, Figure SM 3 to Figure SM 7) (Figure 6), and for the LW$_\downarrow$ across the day (nBE ~2%). The LW$_{Out,Wall}$ nBE is < 0.1%, and the nBE for LW*$_{Wall}$ is 8-11%. The nBE is less when

$T_{Wall}$ is warmer (i.e., middle of day). The larger error in LW*$_{Wall}$ is caused by a small net flux as LW$_{In,Wall}$ and LW$_{Out,Wall}$ cancel each other thus small errors result in large nBE.

SPARTACUS-Urban slightly underestimates LW$_{In,Wall}$ and LW$_{Out,Wall}$ (Figure 6) at the base of the canopy, therefore LW*$_{Wall}$ is slightly overestimated. SPARTACUS-Urban overestimates LW$_{In,Roof}$ below $\overline{H}$. With just one $T_{Roof}$ per time interval,

LW$_{Out,Roof}$ error is small (nBE ~3%), causing underestimates of LW*$_{Roof}$ and larger nBE (5.5 to 8.5 %).

Across the multiple cases with differences in facet $T$ (e.g., magnitude of $T_{Roof} > T_{Wall}$), the agreement is consistent between the two models. These differences may have arisen due to the geometry assumptions in SPARTACUS-Urban or the wall temperature averaging, but despite this, their magnitudes remain low.





**Figure 6:** Longwave fluxes (LW) for a 2 km x 2km domain in central London (Figure 1) simulated with SPARTACUS-Urban (green) and DART (purple) with an emissivity of 0.93 at 5:45 UTC on the 27[th] August 2017 with single facet T: (a) downwelling clear air flux ($LW_\downarrow$), (b) upwelling clear air flux ($LW_\uparrow$), (c, d, e) wall interception, outgoing and net flux ($LW_{In,Wall}$, $LW_{Out,Wall}$, $LW*_{Wall}$), (f, g, h) roof interception, outgoing and net flux ($LW_{In,Roof}$, $LW_{Out,Roof}$, $LW*_{Roof}$).

**Table 2:** Evaluation of SPARTACUS-Urban (cf. DART) for a domain in central London on an August day, for facets prescribed a single surface temperature. Upwelling and downwelling clear-air fluxes ($LW_\downarrow$, $LW_\uparrow$), and the total outgoing and net flux into each urban facet (wall, roof, ground, e.g., $LW_{Out,Wall}$, $LW*_{Wall}$), assessed using the normalised bias error (nBE, Eq. 8).

| Time | $LW_\downarrow$, z = 1 | | $LW_\uparrow$, z = $H_{max}$ | | $LW*_{Wall}$ | $LW*_{Roof}$ | $LW*_{Ground}$ | $LW_{Out,Wall}$ | $LW_{Out,Roof}$ | $LW_{Out,Ground}$ |
|---|---|---|---|---|---|---|---|---|---|---|
| (UTC) | DART | nBE (%) | DART | nBE (%) | nBE (%) | nBE (%) | nBE (%) | nBE (%) | nBE (%) | nBE (%) |
| 5:45 | 10.5 | 2.2 | 26.6 | 0.47 | 11 | -8.2 | -3.3 | 0.047 | -3.3 | -0.24 |
| 7:45 | 10.9 | 2.2 | 28.9 | 0.19 | 9.8 | -6.9 | -3.1 | 0.023 | -3.1 | -0.24 |
| 9:45 | 11.3 | 2.3 | 32.0 | -0.099 | 8.5 | -6.0 | -2.9 | 0.0073 | -2.9 | -0.24 |
| 11:45 | 11.6 | 2.4 | 33.7 | -0.18 | 8.5 | -5.8 | -2.7 | 0.0052 | -2.7 | -0.24 |
| 13:45 | 11.8 | 2.4 | 34.7 | -0.27 | 8.2 | -5.5 | -2.7 | -0.0043 | -2.7 | -0.24 |



| 17:45 | 11.6 | 2.4 | 31.2 | 0.20 | 9.9 | -6.9 | -3.3 | 0.029 | -3.3 | -0.23 |
|---|---|---|---|---|---|---|---|---|---|---|
| 19:45 | 11.3 | 2.2 | 29.1 | 0.40 | 11 | -7.9 | -3.1 | 0.047 | -3.1 | -0.24 |
| 21:45 | 11.2 | 2.2 | 28.4 | 0.45 | 11 | -8.2 | -3.2 | 0.047 | -3.2 | -0.24 |

### 4.3 Comparison of SPARTACUS-Urban and DART: Varying facet temperature with solar irradiance

Second, we compare the two models when facets are prescribed a $T$ range. Here, SPARTACUS-Urban has good agreement with DART for $LW_\downarrow$ at the base of the canopy (nBE 1.7- 2.9%), and at the top of the canopy, for all times (Table 2, Figure 7, Figure 8, Figure SM 8 to Figure SM 12). There are some disagreements towards the centre of the canopy, ~10 – 40 m, at all times, where SPARTACUS-Urban overestimates the $LW_\downarrow$. There is also good agreement in $LW_\uparrow$ up to ~40 m. SPARTACUS-Urban has good agreement (nBE < 0.5%) at the start and end of the day when there is a small range in facet

$T$ (Figure 5), and so temperature averaging (i.e., wall orientation) has little impact. The nBE in $LW_\uparrow$ is poorest in middle of the day (11:45 UTC – 14:45 UTC) when the facets have a large range in temperature but is still < 2.5 %.

The largest errors occur in the LW roof fluxes. The $LW_{In,Roof}$ is always overestimated by SPARTACUS-Urban below the $\overline{H}$ (as in Sect. 4.2). $LW_{Out,Roof}$ is similar to DART (nBE ~ 3%), suggesting the $T_{Sun,Roof}$ and $T_{Sh,Roof}$ averaging method provides a

good approximation to DART. The $LW^*_{Roof}$ is underestimated in SPARTACUS-Urban below $\overline{H}$, with nBE 6 – 8 %. These differences could be associated with SPARTACUS-Urban building height having 1 m resolution whereas DART's roof fluxes are aggregated to each voxel top. Despite this, the vertical profiles of $LW_{Roof}$ fluxes in SPARTACUS-Urban and DART are still close (Figure 7d, h, l).

The SPARTACUS-Urban LW wall fluxes generally compare well to DART. There are slight differences in the $LW_{In,Wall}$ close to the surface, which is likely associated with internal building walls being removed (Sect. 3.1). The nBE in $LW_{Out,Wall}$ is ~8% throughout the day, for all surface temperature configurations. The nBE in $LW^*_{Wall}$ varies from 0 – 10% through the day, with it smallest (11:45 UTC – 14:45 UTC) when there is the largest $T_{Wall}$ variation (Figure 3). The good agreement in $LW_{Out,Ground}$, suggests the averaging method for sunlit and shaded temperatures performs well. SPARTACUS-Urban

underestimates $LW^*_{Ground}$ but with low nBE (2 - 5 %).





**Figure 7: As Figure 6, but for facet temperatures prescribed based on SW simulations at 13:45 UTC. DART simulations use a full temperature profile.**





**Figure 8: As Figure 7, but for 17:45 UTC**

**Table 3: As Table 2, but facets T profile based on SW simulations. DART full temperature profile,**

| Time (UTC) | $LW_\downarrow$, $z=1$ | | $LW_\uparrow$, $z=H_{max}$ | | $LW^*_{Wall}$ | $LW^*_{Roof}$ | $LW^*_{Ground}$ | $LW_{Out,Wall}$ | $LW_{Out,Roof}$ | $LW_{Out,Ground}$ |
|---|---|---|---|---|---|---|---|---|---|---|
| | DART | $nBE$ (%) | DART | nBE (%) | nBE (%) | nBE (%) | nBE (%) | nBE (%) | nBE (%) | nBE (%) |
| 7:45 | 10.8 | 1.9 | 29.1 | -0.31 | 8.0 | -7.3 | -3.3 | 0.047 | -3.3 | -0.23 |
| 9:45 | 11.3 | 1.7 | 33.9 | -2.0 | 1.7 | -7.3 | -3.1 | -0.38 | -3.1 | -0.42 |
| 11:45 | 11.6 | 2.7 | 37.2 | -2.2 | 4.0 | -6.6 | -2.0 | -1.6 | -2.0 | -0.28 |
| 12:45 | 11.7 | 2.2 | 37.9 | -2.5 | 0.62 | -6.8 | -4.8 | -1.5 | -4.8 | -0.923 |
| 13:45 | 11.8 | 2.3 | 37.6 | -2.4 | 0.13 | -6.9 | -3.0 | -1.7 | -3.0 | -0.48 |
| 14:45 | 11.8 | 2.9 | 37.3 | -2.3 | 5.2 | -7.7 | -2.0 | -1.7 | -2.0 | -0.032 |
| 17:45 | 11.5 | 2.4 | 31.2 | -0.15 | 10 | -7.8 | -5.0 | -2.0 | -5.0 | -0.70 |





## 4.4 Impact of surface temperature prescription in SPARTACUS-Urban

Third, as  SPARTACUS-Urban performs well for both temperature scenarios (Sect. 4.2-4.3), we compare using weighted
average vertical profiles of $T_{Wall}$ and $T_{Roof}$ (as Eq. 7, for Harman). This ensures the average emission is the same for each
simulation, allowing both simulations to be compared.

There are negligible differences between the $LW_\uparrow$ and $LW_\downarrow$ within the canopy, for both simulations (Figure 9). As the
geometry is identical between simulations, the $LW_{In,Roof}$ and $LW_{In,Wall}$ are also the same. The nBE in the $LW_{Out,Roof}$ and
$LW_{Out,Wall}$ are small ( $< -0.2\%$), but larger in the $LW_{Out,Ground}$ (nBE $< 4\%$) (Table 4, Figure SM 13). The largest nBE are in the
$LW^*_{Wall}$ (nBE $< -3\%$) and $LW^*_{Ground}$ (nBE $< 4.8\%$). The $LW_{Out,Wall}$ switches from an overestimate to underestimate in the
single $T$ simulation at ~12 m, corresponding to where the single wall temperature overestimates, and then underestimates, the
$T$ profile. This then impacts the $LW^*_{Wall}$ profile. These changes in wall and roof temperature profiles mimic the cumulative
profiles in the wall and roof fraction (Figure SM 2).







**Figure 9: SPARTACUS-Urban simulations for 13:45 UTC with a single ($T_{,Single}$) and range ($T_{Profile}$) facet $T$.**

**Table 4: As Table 2, but between two SPARTACUS-Urban simulations with a single facet $T$ ($T_{Single}$) and facet $T$ range ($T_{Profile}$).**

| Time | LW$_\downarrow$, $z = 1$ | | LW$_\uparrow$, $z = H_{max}$ | | LW*$_{Wall}$ | LW*$_{Roof}$ | LW*$_{Ground}$ | LW$_{Out,Wall}$ | LW$_{Out,Roof}$ | LW$_{Out,Ground}$ |
|---|---|---|---|---|---|---|---|---|---|---|
| | $T_{Profile}$ | $nBE$ (%) | $T_{Profile}$ | nBE (%) | nBE (%) | nBE (%) | nBE (%) | nBE (%) | nBE (%) | nBE (%) |
| 7:45 | 11.0 | 0 | 29.0 | 0 | -1.7 | 0.036 | 2.9 | 0.031 | 2.9 | 1.0 |
| 9:45 | 11.5 | 0 | 33.2 | 0 | -1.9 | 0.063 | 1.2 | -0.017 | 1.2 | 0.65 |
| 11:45 | 11.9 | 0 | 36.4 | 0 | 0.43 | -0.072 | -4.1 | -0.12 | -4.1 | -1.2 |
| 13:45 | 12.0 | 0 | 36.7 | 0 | -1.4 | 0.0045 | -0.54 | -0.11 | -3.7 | -1.1 |
| 17:45 | 11.7 | 0 | 31.1 | 0 | -3.0 | 0.29 | 4.8 | -0.054 | -0.54 | 0.067 |




## 5 Comparison with the Harman et al. (2004) approach

Finally, SPARTACUS-Urban, DART and Harman et al. (2004) are applied to a case with an infinitely long canyon surrounded by buildings of equal height. The temperature configurations are used with the area-weighted SPARTACUS-Urban temperature profiles used in Harman approach (Eq. 7). For the more realistic temperature configurations,
SPARTACUS-Urban single-layer and the Harman approach have similar run-times (Table 5). This increases by a factor of $10^2$ s when realistic geometry is used in SPARTACUS-Urban. The full-temperature DART runs are $10^7$ s slower than the most complex SPARTACUS-Urban simulations.

For simulations with each facet having a single surface temperature (cf. temperature profile), Harman et al. (2004) agrees
better to DART $LW_\uparrow$ at the top-of canopy ($H_{max}$), with 5:45 UTC approximately equal (Figure 10). The poorest Harman agreement to DART is for $LW_{In,Roof}$ and $LW^*_{Wall}$. Although at 5:45 UTC, the nBE $LW^*_{Wall}$ is approximately the same for SPARTACUS-Urban and Harman (Figure 10). This may be because no walls exist above $\overline{H}$, so roofs cannot intercept radiation from above, leading to an underestimate in $LW_{In,Roof}$. When DART has a $T$ range, the Harman performance is similar to the single facet $T$ simulations (Figure 11). However, the nBE are generally higher, except for the $LW^*_{Roof}$ and the
$LW_{Wall}$ fluxes (e.g., 13:45).

Generally, SPARTACUS-Urban agrees more closely to DART than Harman et al. (2004). In the varied facet $T$ simulations, SPARTACUS-Urban and Harman approach are similar for $LW_\uparrow$, and $LW_{In,Roof}$, with nBE < 3%. The two models are similar for $LW_{Out,Ground}$ and $LW_{Out,Wall}$ throughout the day, with the smallest nBE (Figure SM 14, Figure SM 15). Largest differences
are seen in the $LW^*_{Ground}$ (SPARTACUS nBE 2-5% compared to Harman nBE > 20%), and in the $LW^*_{Wall}$ (SPARTACUS nBE 0-10% cf. 8-16%).



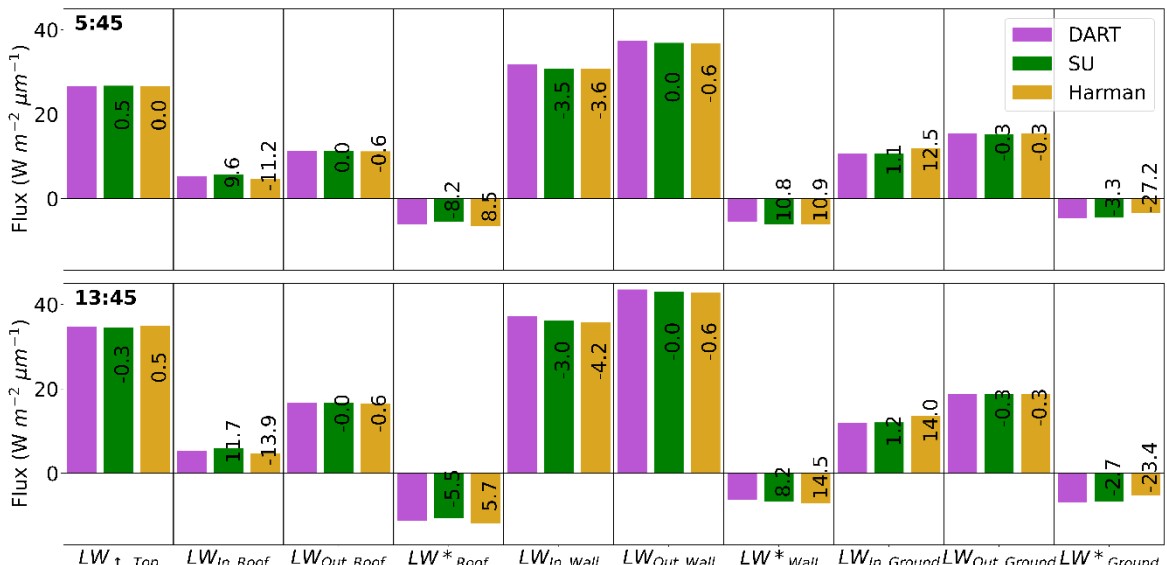

**Figure 10**: **Comparison of DART, SPARTACUS-Urban (SU), and Harman et al. (2004) longwave fluxes for a real-world domain in central London, for an August day. Facet temperatures are isothermal (prescribed as in Sect. 3.3): upwelling clear air flux at the top of the canopy (LW↑), and the roof, wall, and ground total interception, outgoing, and net flux, for two times (rows). Numbers on each bar are the nBE (Eq. 7) between SPARTACUS-Urban/Harman approach and DART**

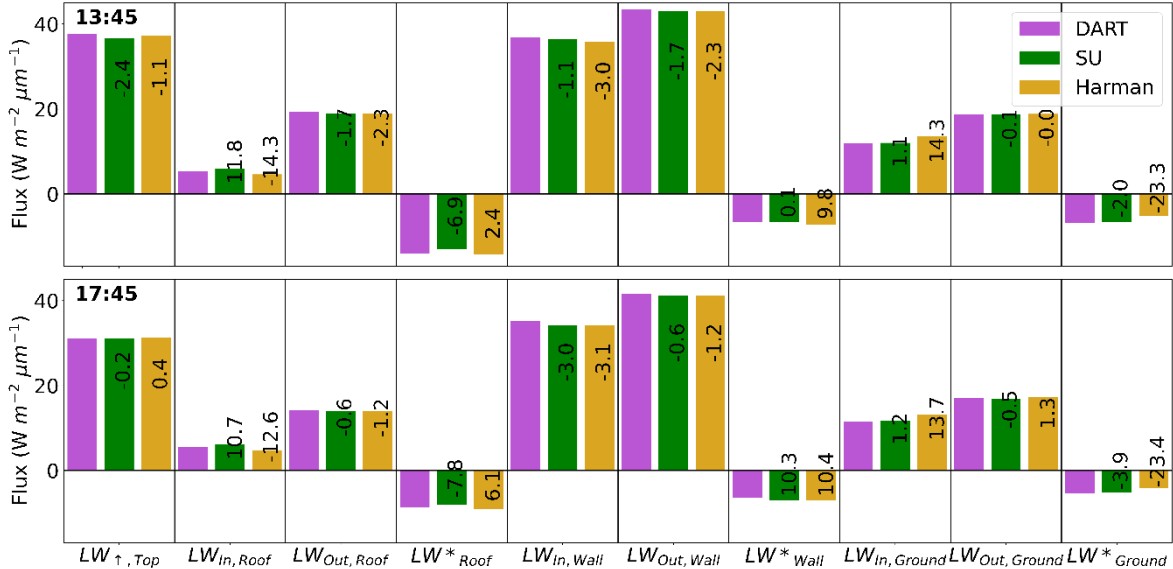

**Figure 11: As Figure 10 but with facets temperature profile prescribed based on SW simulations. DART: full T profile, Harman: area-weighted average of the SPARTACUS-Urban surface temperature profile.**

**Table 5 Absolute run-time of Harman, SPARTACUS-Urban, and DART with the indicated number of vertical layers (n) and diffuse streams per hemisphere (N). Open-source SPARTACUS-Surface version 0.7.3 compiled with gfortran (O3 optimization). Runs undertaken in a single-threaded Linux environment on a dual Xeon E5-2667 v3 processor with 256 GB of RAM. DART version 5.7.5 build number 1126 run in the same Linux environment with 14 parallel threads using 32 CPU.**



| Model | n | N | Time (s) |
|---|---|---|---|
| Harman | 1 | - | $2 \times 10^{-5}$ |
| SPARTACUS-Urban | 1 | 8 | $3 \times 10^{-5}$ |
|  | 6 | 8 | $4 \times 10^{-4}$ |
|  | 151 | 1 | $2 \times 10^{-3}$ |
|  | 151 | 4 | $2 \times 10^{-3}$ |
|  | 151 | 8 | $2 \times 10^{-3}$ |
| DART | 151 | - | $6.6 \times 10^{4}$ |

## 6 Conclusions

Here, the longwave capabilities of the multi-layer radiative transfer model SPARTACUS-Urban are assessed using the explicit radiative transfer model, DART. DART resolves radiative interactions between individual facets of buildings, whereas SPARTACUS-Urban models the mean radiation field with height using building fraction and wall area at each height. Real-world geometry is considered using prescribed surface temperatures ($T$) categorised from urban surface temperature observations measured in London (Morrison et al. 2020, 2021).

Longwave (LW) fluxes are predicted well when one surface $T$ is prescribed per facet type (or sub-facet, e.g., wall orientation). The clear-air upwelling and downwelling fluxes are predicted well, although there is some disagreement in the mid-canopy. SPARTACUS-Urban underestimates (normalised bias errors (nBE) -5.5 – -8.2%) the net LW roof flux, suggesting too much emission from surrounding walls. Errors in this configuration could be from the SPARTACUS-Urban geometry assumptions, or the wall-temperature averaging methods.

Similar agreement is found when facets are prescribed a temperature range based on shortwave simulations. The clear-air fluxes are in good agreement, with nBE less than 3% for all times. The net wall LW is overestimated (nBE up to 10%) at times when of low intra-facet temperature variability (e.g., early morning and evening). Roof interception also is overestimated nearer the ground, leading to an underestimation in the net roof LW. However, all nBE < 11%. This suggests the average $T$ profiles, informed by shortwave geometry are acceptable approximations of the true $T$ field. However, we note the sub-facet wall $T$ range is small, which may differ in different conditions (e.g. atmospheric, geometry).

SPARTACUS-Urban outperforms the frequently used infinite street canyon approach (Harman et al. 2004) (cf. DART). Both are similar if single $T$ facets are used, except for the intercepted roof and net wall LW, when SPARTACUS-Urban is better. When using a facet temperature range the performance for both models is poorer. Harman notably underestimates roof interception, most likely linked to the absence of downward emission from walls higher in the canopy, given all are same height.



The impact of vertically varying *T* is small to SPARTACUS-Urban, with little impact on the net LW fluxes. However, only one summer day in central London is considered, possibly with small variations in wall *T*. In other geometries or climates (e.g., subtropical city with taller buildings), the impact of *T* profile (single, varied) application on the results still needs to be assessed and could be explored in future research.

Overall, this work suggests SPARTACUS-Urban's longwave fluxes agree well relative to the more complex, computationally and data demanding DART model. Alongside the evaluation of SPARTACUS-Urban for shortwave radiation (Stretton et al. 2022), good model performance is shown here, indicating it is suitable for implementing into a multi-layer urban model. Testing is underway with SPARTACUS-Urban coupled to the Surface Urban Energy and Water balance Scheme (SUEWS, Järvi et al. (2011, 2014); Ward et al. (2016); Omidvar et al. (2022), to predict the vertical profile

of fluxes, surface temperatures, and heat stress metrics within the canopy. Such models require high resolution building geometry information (i.e., vertical descriptions of the urban canopy), which are unavailable for most cities. Therefore, to supplement these implementations an assessment should be made on how realistically available data influences model outputs, e.g., vertically distributed fluxes and temperatures.

**Acknowledgments**

The authors acknowledge the funding and support from the Scenario NERC Doctoral Training Partnership Grant, EPSRC 2130186, EPRSC DARE EP/P002331/1, ERC Synergy *urbisphere (855005)*, and Newton Fund/Met Office CSSP China NGC.

**Data availability statement**

The Fortran SPARTACUS-Surface package is available under an open-source license from https://github.com/ecmwf/spartacus-surface. The DART model is available from https://dart.omp.eu. All code and data used for this study are archived at 10.5281/zenodo.6798640

**Competing interests**

The contact author has declared that none of the authors has any competing interests

**Author Contributions**

MS performed the SPARTACUS-Urban simulations, data analysis, and wrote the initial manuscript. WM developed the 3D model, and performed the DART simulations with input from MS. RH is the main author of the SPARTACUS-Surface code,

which was modified by MS. All authors designed the manuscript structure, read, and provided feedback on the manuscript. SG and RH formulated the initial idea. SG obtained for funding to support all except RH.



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
