# Peer review of "Evaluation of vertically resolved longwave radiation in SPARTACUS-Urban 0.7.3 and the sensitivity to urban surface temperatures"

_EGUsphere, 2022_

## Author Comment (AC1)

We would like to thank the reviewers for their comments.

- Our point-by-point responses are in *blue*
- Our proposed changes to the manuscript in *red*. Line numbers (L) are the new ones unless otherwise indicated.

**Reviewer 1:**

The manuscript describes a comparison between a statistically "realistic" 3D model with a more realistic and a parameterised model using a one day one city setting.

- Given that the models are fed in with camera temperature data, and that the "true" values are based on the comparison amongst the models, would it not be better to feed the models with cloud cover and SW incoming data, calculate fluxes and temperatures, and contrast them to the temperature provided by the camera inputs? This approach would provide the possibility of a temperature forecast based on historic cloud and temperature forecasts, which would improve the ability of the modelling approach from just confirming past temperature profiles, to predicting possible extreme temperature scenarios. Given the availability of temperature profiles, I would suggest this modification to the study.

  *This study provides an offline evaluation of the new SPARTACUS-Urban radiative transfer model, against a complex explicit radiative transfer model, DART. As this is conducted offline, both models need to be prescribed with both incoming longwave fluxes and surface temperatures, and then the resultant absorbed and emitted radiative fluxes for each surface type are calculated for those exact input conditions. The comparison here is whether the fluxes for each facet type (wall, roof, ground) from SPARTACUS-Urban agree well with those from the DART model (here classed as the 'truth'), and to make a judgement on the suitability of the SPARTACUS-Urban approach for modelling longwave radiation in urban environments.*

  *This work shows that the SPARTACUS approach performs well against the more complex and 'realistic' DART model, and as such that the SPARTACUS-Urban approach could be integrated within an existing urban climate model. It is in this next stage of work that the surface temperatures should be allowed to evolve, and the longwave fluxes calculated from these, which can then be compared with observed surface temperatures. Without this current evaluation, we would be unsure as to whether SPARTACUS-Urban provides a good prediction of longwave fluxes, and hence whether it should be coupled to existing urban climate or land-surface schemes.*

  *The above has been clarified in the introduction and conclusion of the text.*
  *L80*
  *In this study, the longwave (LW) capabilities are evaluated for the first time. SPARTACUS-Urban's performance is compared to both the explicit scheme DART (Discrete Anisotropic Radiative Transfer, Gastellu-Etchegorry et al. (2015)) and to a common approach used in operational NWP and climate modelling, Harman et al. (2004) (Sect. 2). To examine SPARTACUS-Urban's LW fluxes we simulate an area in*

*central London, with facet temperatures available from thermal camera observations (Morrison et al. 2020, 2021) that can be prescribed with varying levels of complexity for the evaluation (Sect. 3).*

*L423*
*Overall, this offline evaluation suggests SPARTACUS-Urban's longwave fluxes agree well relative to the more complex, computationally and data demanding DART model.*

*L427*
*... to predict the vertical profile of fluxes, surface temperatures, and heat stress metrics within the canopy, with future work including an online evaluation of SPARTACUS-Urban within SUEWS.*

The manuscript requires a major revision in terms of readability. Some suggestions are made here, but there are too many for a referee to make the manuscript readable.

- One major aspect is how can the comparisons extended to LW bands after contrasting only the 10-micrometre wavelength? If this is the best approximation, there should be a section describing how is this approach affecting the evaluation of temperature changes.

  *We agree that a limitation of this study is that only one wavelength (10 μm) has been used, due to the complex nature of the DART simulations. Therefore, there may be some uncertainty if the SPARTACUS-Urban model was applied for different wavelengths. However, we do note that the chosen wavelength is approximately central to that of the longwave infrared band.*
  *Text clarified*
  *L164*
  *Given computational constraints, DART is run for a single wavelength (10 μm). We choose 10 μm, as it is approximately central to the LW infrared band, hence some additional uncertainty arises in SPARTACUS-Urban results for other wavelengths and broadband longwave flux measurements cannot be used.*

- Section 2.2 This paragraph uses too technical terminology that is not explained in the text.

  *The following paragraphs are proposed to replace current text in Section 2.2 of the manuscript to describe the DART radiative transfer model.*
  *L109*
  *The DART (Discrete Anisotropic Radiative Transfer) model (Gastellu-Etchegorry et al. 2015) can simulate variability of radiative exchanges across one SPARTACUS-Urban grid cell in detail using a 3D digital surface model (DSM) with vegetation, buildings and atmosphere. Each voxel (or grid box) size has a user-prescribed resolution. The model domain's elements (e.g., vegetation, buildings) within a voxel can interact with each other. The per-voxel radiative budget products are stored after each numerical iteration. DART scene elements are often represented by flat 'triangles' making up building walls and roofs or leaves on trees. Each triangle has an area, orientation, and optical properties. Alternatively, DART can represent vegetation as 'turbid media' (or volumes filled with randomly distributed infinitely small facets) characterised by an angular distribution and an area volume density.*

*To model the urban LW field in DART, both a 3D building model and a 3D field of surface temperatures are required. The latter can be prescribed based on solar irradiance state (e.g., currently sunlit, shaded). Here, each building's triangles are categorised based on facet type (e.g., roof, wall) and orientation (e.g., west, east) to allow realistic spatial values. As a triangle can have only one temperature, if a triangle covers a whole wall (i.e., vertical building facet) there is no vertical variation.*

- Ln 75: Which two parameters?

  *The two parameters are the building fraction and building edge length at each height level, which are both required as inputs to the SPARTACUS-Urban radiative transfer model. Text clarified.*

  *L74*

  *… with geometry describable by vertical profiles of building plan area and building edge length*

- Ln 37 This sentence needs to be rethought, as the verb "increased" does not match the syntax.

  *An extra 'the' has been removed after the word 'increased'. Modified  L36*

  *The crenulated urban morphology and resultant deep canyons cause an uneven exposure to the sky and an increased surface area available for exchange (cf. rural areas), which increases the SW absorption throughout the day..*

- Ln 49 "need" instead of "need"

  *Modified  L50*

  *These impacts on the radiative and other energy exchanges need to be parameterised within numerical weather prediction (NWP) land surface schemes (Masson 2006).*

- Ln 104 "makes" instead of "make"

  *Modified L105*

  *For this paper, we assume a wavelength of 10 μm (where atmospheric absorption is weak), so the emission rate in SPARTACUS-Urban (and DART) makes use of the Plank function…*

- Ln 124 Full stop missing.

  *Modified  L133*

  *… finds agreement between the two models for the net outward LW flux from the ground and walls when SPARTACUS uses more than 4 streams. Here, the SPARTACUS-Surface software package (see Sect. 4.2 of (Hogan 2019a)) implementation of Harman is used for the simulations.*

**Reviewer 2:**

The paper by Stretton et al. compares the longwave radiation component of SPARTACUS-Urban with the DART model and the model by Harman et al. (2004) for a central London (UK) domain. Different surface temperatures based on observations are prescribed. While the findings of this paper are import for the modelling community and the general approach is sound, the explanation of the derivation of the surface temperatures is confusing and important discussion points are missing. Thus, I recommend reconsideration of this work after major revisions.

Major issues:

1) In which way is DART suited to be the reference of a model evaluation? What are the expected errors of DART itself? The paper mentions an evaluation of DART for vegetation? Was there an evaluation in an urban area? DART was also used as a reference model for the shortwave part of SPARTACUS-URBAN. This does not make it automatically suited for the present longwave part, does it?

> *DART is an explicit radiative transfer model so contains the detailed radiative interactions unlike the simpler radiative transfer models (e.g., SPARTACUS) being evaluated. The following text is added*
> *Text modified.*
> *L123*
> *Given DART is an explicit radiative transfer model it has more detailed radiative interactions than the simpler radiative transfer models (e.g., SPARTACUS, Harman). DART has been evaluated in vegetated areas using thermal infrared observations (Sobrino et al. 2011) and relative to other models in the RAMI intercomparison project (Widlowski et al. 2015). The DART version including buildings (Gastellu-Etchegorry et al. 2015) has not been explicitly evaluated in urban areas, but has been used to assess urban SW and LW radiation and albedo (Chrysoulakis et al. 2018; Landier et al. 2018), variations in urban surface temperatures (Morrison et al. 2020, 2021), and mean radiant temperature (Dissegna et al. 2021), and to assess simpler radiative transfer models (e.g., SPARTACUS-Urban, Stretton et al. (2022)).*

2) SPARTACUS-Urban is a multi-layer urban scheme. The paper misses to introduce more multi-layer schemes, in particular the widely used BEP (Martilli et al. 2002, implemented in WRF) and the schemes based on it. The approaches in the radiation exchange parametrization should be quickly compared. Multi-layer schemes are also not considered in the Discussion/Conclusion part of the paper. In which respect does the results of SPARTACUS-Urban are expected to differ from these more traditional, non-statistical urban schemes? Why should one prefer the new approach compared to the older, street-canyon view-factor based approach? The paper mentions that roof interception is a problem for the single-layer Harman et al. model because of missing radiation from higher walls. Other multi-layer models are able to consider this (e.g. Schubert et al. 2012).

> *We agree the description of multi-layer schemes in the Introduction and Conclusion of our paper is brief, so have added other examples in the text.*
> *Our aim is not to intercompare SPARTACUS-Urban scheme to existing multi-layer modelling approaches, but to first assess its performance using an explicit radiative transfer model. The Harman scheme is included to check that SPARTACUS-Urban*

*performs similarly, or better, than the simpler but commonly used street-canyon approach. Future studies could intercomparison multi-layer (and potentially single-layer) approaches to modelling radiation within urban areas, like the Urban-PLUMBER project (Lipson et al. 2023). This has been added as a suggestion to the conclusion of the paper.*

*L69*

*Some of these features can be addressed by utilising multi-layer radiative transfer models, allowing more nuanced radiative trapping and realistic vertical temperature distributions (e.g. Seoul National University Canopy Model (Ryu and Baik 2012; Ryu et al. 2013), Building Effect Parameterisation (BEP, Martilli et al. (2002); Schubert et al. (2012)), the Town Energy Balance model (TEB, Hamdi and Masson (2008)), and SPARTACUS-Urban (Hogan 2019a)). Most assume a canyon geometry, those with varying building heights permitting more realistic inter-building shading (e.g., Schubert et al. (2012)). SPARTACUS-Urban assumes buildings are distributed randomly in the horizontal plane, with geometry describable by vertical profiles of building plan area and building edge length, allowing radiative exchanges simulations fast enough for NWP accounting for atmospheric absorption, emission, and scattering between buildings. The approach provides a more accurate description of radiation exchange than single layer street-canyon approaches (Hogan 2019b). The shortwave (SW) simulations for realistic urban domains have good agreement to an explicit radiative transfer model (Stretton et al. 2022).*

*L448*

*Further, comparisons could be made between existing single- and multi-layer urban radiative transfer schemes, such as done in the RAMI intercomparison for vegetation (Widlowski et al. 2015), or urban energy balance intercomparisons (Grimmond et al. 2010, 2011) (Lipson et al. 2023).*

3) The description of the averaging of the observed surface temperatures is not completely clear.

- In 3.2, it is stated that the surfaces were already separated into sunlit and shaded surfaces. Why is this process repeated with shortwave SPARTACUS-urban? Is it because the information of the observations is only known for a sub-area of the full analysis domain? Is it expected that the fractions of sunlit and shaded surfaces is considerably different in the sub-area and in full domain?

  *As DART has "all" the surfaces represented but SPARTACUS-Urban has simplified vertical profile geometry, consistency of description between sunlit and shaded needs to be ensured. It is much more complex to determine all the sunlit and shaded fractions in DART, So, SPARTACUS-Urban shortwave simulations are completed for each time interval, with the correct solar zenith angle, to obtain the sunlit fraction of walls and roofs. Text clarified.*

  *L212*

  *As it is complex to extract the vertical profile of temperature for each surface type from DART, solar zenith angle ($\theta_0$) dependent SW SPARTACUS-Urban simulations are used to estimate the sunlit fraction for the walls ($F_{Sun,Wall,i}$) and roofs $F_{Sun,Roof,i}$) by*

*height interval, and for the ground ($F_{Sun,Ground}$). The shaded fractions are obtained by difference ($F_{Sh,Wall,i} = 1 - F_{Sun,Wall,i}$). The appropriate DART sunlit (shaded) temperatures are assigned to SPARTACUS-Urbans sunlit (shaded) fraction. Similarly, the sunlit and shaded roof temperatures ($T_{Sun,Roof}$, $T_{Sh,Roof}$) are weighted at each height by the appropriate sunlit and shaded fractions to obtain $T_{Roof,i}$ and at $z=0$ for the ground ($T_{Ground,sun}$, $T_{Ground,sh}$). Thus, enabling SPARTACUS-Urban to capture the horizontal surface temperature variations.*

- \* L198: How is a surface temperature range prescribed in DART?

  *When a surface temperature facet range is used, a DART SW simulation is conducted for that time to determine if the surface (triangles) are sunlit or shaded allowing appropriate facet orientation (e.g., roof, west wall, north wall) 'sunlit' or 'shaded' temperature to have the appropriate minimum or maximum temperature range observed by the thermal cameras (Fig 7, Morrison et al. 2021 assigned. Note, triangles can cover whole walls. Text clarified.*

  *L205*

  *A temperature range can be prescribed in DART allowing sunlit-shaded variations. However, given level of detail of the surface model used (Figure 1) the observed surface temperatures are not directly usable as camera pixels has much higher resolution than the DART triangles. DART SW simulations are used to determine whether each facet triangle is sunlit or shaded, and therefore which temperature (maximum/minimum) range (Figure 3) is assigned by type (e.g., roof, west facing wall, east facing wall). As noted, as DART triangles may have whole wall resolution but only one prescribed temperature.*

- \* L215: Did you know for each observation whether it was sunlit or shaded? Then these values could have been averaged directly before calculating orientation specific values. With orientation-averages, you had to introduce a somewhat arbitrary averaging.

  *The Morrison et al. (2020, 2021) 7 thermal cameras (e.g., Figure 2a, Morrison et al. 2021) pixels are classified by timestep (e.g., Fig 7, Morrison et al. 2021). Here, DART triangle are prescribed temperatures based on facet type, orientation, and sun/shade status, but is limited to only one temperature per triangle per time. The triangles are coarser than the camera pixels. The Morrison et al. (2020, 2021) observation site (420 m x 420 m) is smaller than the full domain used here (2 km x 2 km). To address this we assume the maximum and minimum temperatures observed correspond to the 'sunlit' and 'shaded' temperatures (Error bars below 0, Figure 5).*

  *Modified text (prior two bullet points), and existing text in Sect. 3.1 address this.*

- \* I do not understand what was studied in 4.4. Did SPARTACUS-urban use the temperatures otherwise prescribed to the Harman model? What is the principal difference to the case in 4.2? In the end, it is just prescribed temperature anyway?

  *Section 4.4 examines differences in SPARTACUS-Urban predicted fluxes when wall and roofs surface temperatures are prescribed as (1) a single (section 4.2) or (2) a profile (section 4.3) of surface temperature. As the results (most evident in the nBE,*

*Table 4) show differences occur between these two assumptions, indicating a temperature profile should be used, with the consequence that SPARTACUS-Urban when coupled to an urban climate model to obtain other fluxes (e.g., storage heat flux) needs to be vertically distributed. Text clarified.*
*L335*

*As SPARTACUS-Urban performs well (cf. DART) for both temperature scenarios (Sect. 4.2, 4.3), we examine differences between using a single facet temperatures (Sect. 4.2) or a profile ($T_{Profile}$, Sect. 4.3). To ensure the average emission is the same in each, the single temperature SPARTACUS-Urban simulations use weighted mean vertical profiles of $T_{Wall}$ and $T_{Roof}$ (Eq. 7, as for Harman).*

Minor issues:

- 4) The paper is only about the urban part of SPARTACUS-Surface. Thus, I recommend using SPARTACUS-urban instead of SPARTACUS-Surface in the title.
  *Title modified*
  *L1*
  *Evaluation of vertically resolved longwave radiation in SPARTACUS-Urban 0.7.3 and the sensitivity to urban surface temperatures*

- 5) I recommend giving the computational times in Table 5 in relative units, for example in units of time needed for the Harman model. This absolute value can be given in the caption of the table. The factors in L336 should be without the unit "s".
  *Table modified and text changed*
  *L362*
  *This increases by a factor of $10^2$ when realistic geometries are used in SPARTACUS-Urban. The full-temperature DART runs are a factor of $10^7$ slower than the most complex SPARTACUS-Urban simulations.*
  *Table 5*

| Model | n | N | Time (s) | Time relative to Harman |
|---|---|---|---|---|
| Harman | 1 | - | $2 \times 10^{-5}$ | - |
| SPARTACUS-Urban | 1 | 8 | $3 \times 10^{-5}$ | 1.5 |
| | 6 | 8 | $4 \times 10^{-4}$ | 20 |
| | 151 | 1 | $2 \times 10^{-3}$ | 100 |
| | 151 | 4 | $2 \times 10^{-3}$ | 100 |
| | 151 | 8 | $2 \times 10^{-3}$ | 100 |
| DART | 151 | - | $6.6 \times 10^4$ | $3.3 \times 10^9$ |

- 6) L141: What does this sentence mean? How are the 25th and 75th percentile used?
  *This sentence describes how the building height is determined for each building from the DEM and the DSM. Text clarified*
  *L149*
  *To simplify buildings so they have flat both roofs and walls, for each building the 25th percentile of the DEM and 75th percentile of the DSM heights are used.*

- 7) The abstract does not give results of the comparison with the Harman model.
  *Abstract modified*

[revised manuscript text omitted]

- *\* Add what we see for Figure SM1. The y axis misses a label.*
  *Label added to Figure SM 1 y-axis. Figure caption expanded.*
  *Figure SM 1*
  *Demonstration of the $T_{Wall}$ averaging method used for SPARTACUS-Urban simulations (Section 3.3). Wall types are weighted using (a) solar azimuth angle for each time using Eq. 4 and Eq. 5, resulting in (b-d) sunlit fraction profiles per time step for weighing sunlit-shaded temperatures (Figure 3) profiles (Figure 5). (a) Azimuth angle of times (red) scatter used in other panels.*

[Figure]

- *\* Add details to Figure SM2.*
  *Text clarified and figure edited.*
  *Figure SM 2*

*Vertical profiles derived for SPARTACUS-Urban grid-cell from a 2 km x 2 km DART domain in central London of (a) fractional wall (black) and roof (blue) area and (b) cumulative fraction, with mean building height (grey dash lines), and heights when areal weighted cumulative fraction is 0.50 (dotted blue and grey lines).*

[Figure]

**References:**

[revised manuscript text omitted]

---

## Author Response (AR2)

We would like to thank the editor for their comment and decision.

- Our point-by-point responses are in *blue*

**Editor:**

- I am happy to accept your paper subject to one technical adaptation:
L362

This increases by a factor of  $10^2$  when realistic geometries are used in SPARTACUS-Urban. The full-temperature DART runs are a factor of  $10^7$  slower than the most complex SPARTACUS-Urban simulations.

Would you please add on what machine you are running and if that is wall time, cpu time, node-seconds, etc.

- *The details of the runs compared between DART/SPARTACUS/Harman are given in Table 5. The full details of the computation and machine are given in the caption to Table 5.*

**Table 1** Absolute run-time of Harman (Sect. Error! Reference source not found.), SPARTACUS-Urban (open-source version 0.7.3 compiled with gfortran, O3 optimization), and DART (version 5.7.5 build number 1126) for simulations with n vertical layers, and N diffuse streams per hemisphere. All runs undertaken in a Linux environment on a dual Xeon E5-2667 v3 processor with 256 GB of RAM, with a single-thread for Harman and SPARTACUS-Urban, but for DART 14 parallel threads using 32 CPU.

| Model           | n   | N | Time (s)           | Time relative to Harman |
|-----------------|-----|---|--------------------|-------------------------|
| Harman          | 1   | - | $2 \times 10^{-5}$ | -                       |
| SPARTACUS-Urban | 1   | 8 | $3 \times 10^{-5}$ | 1.5                     |
|                 | 6   | 8 | $4 \times 10^{-4}$ | 20                      |
|                 | 151 | 1 | $2 \times 10^{-3}$ | 100                     |
|                 | 151 | 4 | $2 \times 10^{-3}$ | 100                     |
|                 | 151 | 8 | $2 \times 10^{-3}$ | 100                     |
| DART            | 151 | - | $6.6 \times 10^4$  | $3.3 \times 10^9$       |